# Recent Synthetic Advances on the Use of Diazo Compounds Catalyzed by Metalloporphyrins

**DOI:** 10.3390/molecules28186683

**Published:** 2023-09-18

**Authors:** Mário M. Q. Simões, José A. S. Cavaleiro, Vitor F. Ferreira

**Affiliations:** 1Department of Chemistry & LAQV-REQUIMTE, University of Aveiro, 3810-193 Aveiro, Portugal; msimoes@ua.pt (M.M.Q.S.); jcavaleiro@ua.pt (J.A.S.C.); 2Departamento de Tecnologia Farmacêutica Química, Universidade Federal Fluminense, Niterói 24241-002, RJ, Brazil

**Keywords:** diazo compounds, porphyrins, carbenes, metallocarbenes, catalytic reactions

## Abstract

Diazo compounds are organic substances that are often used as precursors in organic synthesis like cyclization reactions, olefinations, cyclopropanations, cyclopropenations, rearrangements, and carbene or metallocarbene insertions into C−H, N−H, O−H, S−H, and Si−H bonds. Typically, reactions from diazo compounds are catalyzed by transition metals with various ligands that modulate the capacity and selectivity of the catalyst. These ligands can modify and enhance chemoselectivity in the substrate, regioselectivity and enantioselectivity by reflecting these preferences in the products. Porphyrins have been used as catalysts in several important reactions for organic synthesis and also in several medicinal applications. In the chemistry of diazo compounds, porphyrins are very efficient as catalysts when complexed with low-cost metals (e.g., Fe and Co) and, therefore, in recent years, this has been the subject of significant research. This review will summarize the advances in the studies involving the field of diazo compounds catalyzed by metalloporphyrins (M−Porph, M = Fe, Ru, Os, Co, Rh, Ir) in the last five years to provide a clear overview and possible opportunities for future applications. Also, at the end of this review, the properties of artificial metalloenzymes and hemoproteins as biocatalysts for a broad range of applications, namely those concerning carbene-transfer reactions, will be considered.

## 1. Introduction

Diazo compounds have been known since the synthesis of ethyl diazoacetate (EDA), which was the first aliphatic organic substance to have a diazo group, in 1883 by Curtius [1,2,3]. Since then, they have fascinated organic chemists due to their synthetic versatility through many bond-forming reactions (C−C, C=C, C−O, C−N, C−S, etc.), including the formation of carbocyclic and heterocyclic compounds. Reactions with diazo compounds involve the formation of intermediate carbenes (R_1_R_2_C:) that can be produced under different reaction conditions such as heating, light irradiation, or Lewis and Brönsted acids decomposition. Metal–carbene complexes or metallocarbenes (R_1_R_2_C=M) are carbenes bonded to a metal. The first of these metallocarbenes was discovered by Schrock [Ta(CH_2_tBu)_3_(CHtBu)] [4], and since then, it has expanded to many other metals, mainly to complexes of the early transition metals from groups 4–6, with the most diverse types of ligands and became important reactive synthons that were used in the most diverse syntheses. For a long time, and up to the present day, these compounds continue to be of interest for organic synthesis and have been reviewed by many researchers in scientific journals and books, mainly in using more efficient and selective catalysts [5,6,7,8]. The scope of reactions involving carbenes and metallocarbenes represents a significantly broader and more versatile approach in organic synthesis compared to reactions with diazo compounds [9,10,11,12]. Carbenes are species containing only two groups covalently bonded to a carbon and two free non-bonding electrons that can be spin-paired in the same orbital (singlet state) or unpaired in two separate orbitals (triplet state) [13]. Therefore, they are electrically neutral species that can simultaneously form two bonds. Metal–carbene complexes are species made up of carbenes that are covalently bonded to a metal, usually with expensive transition metals (M = Ru, Os, Co, Rh, Ir) [14,15]; however, there are several works with iron, an abundant metal, which is cheap, environmentally benign, and with low toxicity [16,17,18,19,20,21,22]. The chemical behavior of carbenes and metal–carbene complexes differ in reactivity and selectivity. Figure 1 shows the reaction of carbenes and metal–carbene complexes with olefins, forming two C−C bonds in one step leading to cyclopropanes. Depending on the specific combination of initial diazo compounds and olefins, this reaction can yield a vast array of organic compounds. Moreover, beyond intermolecular reactions involving diazo compounds, they also demonstrate the potential to engage in diverse intramolecular reactions, including cyclopropanations, C−H, O−H, S−H, N−H bond insertions, and aromatic cycloaddition reactions.

Diazo compounds are used in syntheses that produce drugs, agrochemicals, pesticides, and derivatives that can be used to prepare other substances. The chemistry of these compounds is widely recognized precisely for having significant importance in total or partial syntheses. They are the best synthetic building blocks for required target molecules due to their high reactivity, versatility, multiplicity of reactions and preparation methods. As amphiphilic reagents, they can undergo geminal difunctionalization by replacing the corresponding diazo function with two new substituents, which are favored by the release of nitrogen [23]. In Figure 2, some reactions are highlighted, among many others, that can be carried out with diazo compounds, under heating, catalysis with metals, Lewis and Bronsted acid catalysis, and photochemistry to form carbocycles and heterocycles. In Figure 2, R^1^ can represent alkyl, aryl, H, CHO, CO_2_R, and R^2^ can also be alkyl, aryl, H, CHO, CO_2_R. Additionally, R^1^ can be different or equal to R^2^ in Figure 2.

Discovered in the 1970s, transition metal carbene radicals (R_1_R_2_^•^C=M) turned out to be fundamental in contemporary catalysis, usually referred to as metalloradical catalysis (MRC). MRC flourishing applications led to the synthesis of countless organic compounds, from simple cyclopropanes to trickier eight-membered rings. So far, the emphasis has mostly been put on Co(II) based systems (R_1_R_2_^•^C=Co), which are highly appropriate for catalytic carbene transfer reactions [24,25,26,27].

## 2. Porphyrin-Based Catalysts

Porphyrins are macrocyclic organic compounds with a general structure with a tetrapyrrole macrocycle as the central part, the four pyrrolic subunits connected by methine bridges. The general skeleton of porphyrins has been extensively modified in several positions to adjust their reactivity, envisaging multiple applications, including catalysis (Figure 1).

The outer part of these heterocyclic macrocycles can present several topologies and numerous substitutions, whereas the central core has rigid planarity in several macrocycles (Figure 1). These molecules present structural arrangements that form cavities, which have enough space to accommodate several metal ions bound to the nitrogens in the central cavity. They perform important biochemical functions (e.g., myoglobin, hemoglobin, cytochromes P_450_) and have been used in medicinal chemistry, in catalysis for various reactions, for the inactivation of microorganisms, as detection probes and materials science. It has been known for many years that metalloporphyrins are important metal catalysts for various transformations. These metal-containing macrocycles react with diazo compounds and induce the formation of carbenes or metallocarbenes, which are helpful in various synthetic transformations. In this sense, diazo compounds are excellent reactants for interacting with porphyrins in inter- or intramolecular reactions and structural modifications in the outer part of the macrocyclic ring. As metalloporphyrins have a metal ion in the center of each macrocycle, they can form metallocarbene intermediates (mainly from diazo compounds) that can catalyze various reactions, such as inter- or intramolecular cyclopropanations, cyclopropenation reactions, carbene carbonylation, olefination of carbonyl compounds, formation of C−C, C−N, C−O, C−Si, C−S bonds, intramolecular Buchner reaction, among others [10,11,17,28,29,30,31,32].

## 3. Reactions through Unusual Strategies

Nitrogen heterocyclic chemical entities are present in many drugs used in clinical medicine. Any efficient method focusing on their efficient preparation is very important in Medicinal Chemistry. One of these prominent *N*-heterocycles is piperidine, which is present as a pharmacophore group in several drugs in the therapeutic arsenal. There are many attractive synthetic routes for the synthesis of piperidines and their analogues.

Based on recent syntheses of 5-membered *N*−heterocycles by metalloradical cyclization catalyzed by Co(II) porphyrins [33], de Bruin and collaborators reported a robust method with high global yields for the synthesis of piperidines directly starting from γ-amino substituted linear aldehydes. The reaction occurs by ring-closing C−C bond formation after the in-situ formation of *N*−tosylhydrazones from aldehydes. This is followed by in situ deprotonation of the hydrazones to afford the corresponding diazo compounds (Figure 3). In this reaction, there is also the formation of linear alkenes as secondary products in small amounts [34]. It is crucial to emphasize that in the absence of the Co complex of *meso*-tetraphenyl porphyrin (CoTPP), only the diazo compound would be formed, and the cyclization product would not be obtained.

A unique route to prepare eight-membered rings was also developed by de Bruin and collaborators, based on metalloradical catalyzed reactions, to construct a series of novel dibenzocyclooctenes, monobenzocyclooctadienes and 8-membered heterocyclic enol ethers, which have been synthesized in good to excellent yields and with excellent substituent tolerance using Co(TPP) as a catalyst, thus producing cobalt(III)-carbene radicals as intermediates [35,36,37]. The metalloradical activation of *N*-tosyl hydrazones with Co(TPP) offered a novel route to build a series of dibenzocyclooctenes, in good to excellent yields through selective C_carbene_−C_aryl_ cyclization (Figure 4A).

Moreover, various aromatic substituents were tolerated on both benzene rings. The proposed mechanism consists of intramolecular hydrogen atom transfer (HAT) to Co(III)–carbene radical intermediates, followed by the dissociation of an ortho-quinodimethane that undergoes 8π cyclization. The presence of radical-type intermediates was confirmed by trapping experiments [35]. The protocol was extended to the synthesis of the novel monobenzocyclooctadienes that can be synthesized by ring-closure of *o*-aryl aldehydes containing bis-allylic C−H bonds. A different mechanism is present here, suggesting that ring-closure to the monobenzocyclooctadienes involves a direct radical-rebound step within the coordination sphere of cobalt, thus tolerating enantioselective formation of chiral monobenzocyclooctadienes (Figure 4B,C) [36].

## 4. X−H Functionalization Catalyzed by Metalloporphyrins

The functionalization of C−H bonds with diazo compounds catalyzed by high-cost transition metals has been explored for many years, mainly with Rh, and many examples can be found in the literature [8,10,38,39]. However, developing efficient and sustainable reactions with non-precious metals is still of little use in processes involving diazo compounds; in this sense, iron, being an abundant and low-cost metal, is an excellent candidate for catalysis. Hock et al. [40] developed highly efficient functionalization reactions on C−H bonds by diazoacetonitrile with *N*-heterocycles catalyzed by the iron tetraphenylporphyrin Fe(TPP)Cl (Figure 5). This protocol makes it possible to prepare important precursors of indole and indazole heterocycles. The authors investigated the possibility of this reaction going through a mechanism via free radicals, conducting the reaction in the presence of TEMPO. This classic radical scavenger completely inhibited the C−H functionalization reactions.

Fe(TPP)Cl also catalyzes the insertion of benzylic carbenes, generated in situ from *N*-tosylhydrazones derived from different benzaldehydes, into X−H (X = Si, Sn, Ge). Toluene and NaH were shown to be the best solvent and base, respectively. Silanes with tertiary, secondary, and primary Si−H bonds afforded the corresponding insertion products in moderate to high yields, and a stepwise double insertion strategy was developed to synthesize unsymmetrical tetrasubstituted silanes. Moreover, this reaction could be extended to Sn−H and Ge−H bonds, affording the insertion products in good to high yields (Figure 6) [41].

Recently, the synthesis and characterization of several donor-acceptor iron porphyrin carbene complexes obtained from an iron porphyrin and the corresponding donor-acceptor diazo compounds were reported. Furthermore, the crystal structure of a donor-acceptor iron porphyrin carbene complex derived from a morpholine-substituted diazo amide was obtained. The carbene transfer reactivity of the iron porphyrin carbenes was studied for the N−H insertion reaction with aniline or morpholine (Figure 7). However, the morpholine-derived amide complex could only react with aniline to deliver the corresponding N−H insertion product with very poor efficiency. Hence, iron porphyrin carbenes were identified as the real intermediates for iron porphyrin-catalyzed carbene transfer reactions from donor-acceptor diazo compounds [42].

Phenols and naphthols are present in many natural products, dyes, pharmaceuticals, functional polymers, etc. These versatile building blocks readily engage in diverse transformations through conventional reactions and exhibit the unique ability to undergo insertion into vicinal C−H bonds adjacent to OH groups. The importance of these substances stimulates research into new direct functionalization strategies on the C−H bond in a chemoselective manner in the presence of other functionalities. For example, the chemo- and site-selective C−H functionalization of unprotected phenols and naphthols. However, this type of functionalization presents considerable challenges. Yu et al. developed a novel iron-catalyzed chemo and regioselective *ortho* C−H bond functionalization of phenols and naphthols with diazoesters. In this transformation, several iron porphyrins were used as catalysts, the best results being obtained for *tetrakis*-(2,4,6-trifluorophenyl)porphyrin chloride Fe(2,4,6-TFPP)Cl (Figure 8) [43].

Metalloporphyrin dialkylcarbene or bis(dialkylcarbene) complexes prepared from diazo compounds as carbene sources are rare in the literature and with little structural information by NMR spectroscopy and X-ray crystallography. They are generally difficult to use in synthetic methodologies due to their low stability. Furthermore, dialkylcarbene metalloporphyrin species may be prone to undergo a 1,2-hydride migration side reaction. However, Che and collaborators [44] were able to solve these problems by synthesizing stable porphyrin complexes of mono and bis(dialkylcarbene) group 8 metals (Fe, Ru, Os) with the linker adamantane (generated from photolysis of aziadamantane, a diazirine compound, as the carbene source). The steric effect of the bulky ligand on the porphyrin macrocycle makes the Fe−porphyrin complex an effective, active, and robust catalyst for the intermolecular transfer of diarylcarbene in reactions including cyclopropanation and S−H, N−H, O−H, and C(sp^3^)-H insertion. The authors synthesized several dialkylcarbene carbene complexes, but for this article, Fe^II^(TPFPP) = adamantane (bright red solid) is highlighted in Figure 9. Its preparation involves treating Fe^II^(TPFPP) with aziadamantane at room temperature under UV irradiation (365 nm) for 15 min (Figure 9).

The same research group described the Ir(III) porphyrin-catalyzed intermolecular C(sp^3^)−H insertion reaction of a quinoid carbene (QC), which showed to be efficient for the arylation of activated hydrocarbons such as 1,4-cyclohexadienes, thus giving functionalized phenol moieties anchored onto 1,4-cyclohexadienes (Figure 10). Moreover, mechanistic studies point out a radical mechanism for these insertion reactions, this methodology being enabled by the hydrogen-atom transfer (HAT) reactivity of the Ir(III)-QC intermediate. The system exhibits distinctive regioselectivity, mainly ruled by steric effects since the insertion into primary C−H bonds is favored over secondary and/or tertiary C−H bonds in substituted cyclohexene substrates [45].

Combining metallo- and photocatalysis, Ir(III) porphyrin-based porous MOFs catalyzed intermolecular C(sp^3^)−H insertion reaction under visible light was also reported, allowing the isolation and structural characterization of an Ir(III) porphyrin-carbene species. Different inert substrates, including cyclic (cyclopentane or cyclohexane) and acyclic (pentane, 2,3-dimethylbutane, 3-methylpentane or hexane) gave satisfactory yields both with ethyl diazoacetate (EDA) or with quinoid carbene (QC) (Figure 2) [46].

Based on Ir(TCPP)Cl (TCPP = *tetrakis*(4-carboxyphenyl)porphyrin), the metal-organic framework (MOF), Ir-PMOF-1(Hf), which can resist acid conditions, has been examined in the O−H insertion reaction of carboxylic acids with diazo compounds, results showing that Ir-PMOF-1(Hf) is an efficient heterogeneous catalyst for these reactions (Figure 11). Moreover, Ir-PMOF-1(Hf) maintained its framework after the catalytic reactions and could be recycled and reused for at least ten runs. For other carboxylic acids, the yields range from 57% after 10 min (for 2-hydroxybenzoic acid as substrate) to 91% after 6 min of reaction (for trifluoroacetic acid) [47].

As reported by the same research group, the heterogeneous catalyst Ru-PMOF-1(Hf), based on Ru(TCPP)(CO) (TCPP = *tetrakis*(4-carboxyphenyl)porphyrin), evidenced catalytic efficiency for the N−H insertion reactions of EDA into a range of secondary amines with up to 92% yield (Figure 12). Due to its 3D structure with orthogonal 1D open channels, Ru-PMOF-1(Hf) induces size selectivity, displaying an apparent yield-decreasing tendency along the chain lengthening (yield for diethylamine > dipropylamine > dibutylamine > dipentylamine). Moreover, Ru-PMOF-1(Hf) could be recovered and reused for at least ten runs with negligible loss of catalytic activity [48].

The development of metalloporphyrin-based capsules showed to be a promising strategy to create nanoreactors, real nanoscale chemical environments where chemical transformations can occur, thus changing the reactivity of molecules upon binding inside the cavity, for example, in carbene transfer/insertion reactions. A recent example deals with a Ru(II) porphyrin-based molecular nanoreactor, bearing a stable and inert diphenylcarbene axial ligand as a catalyst in selective N−H carbene insertion reactions, where no signs of dimerization side processes nor double insertions were observed in the rotaxane assembly reaction, hence leading to the quantitative formation of rotaxanes by active-metal-template synthetic methodology. Only the internal axial position of the Ru(II) catalyst is available for activation of the substrates, explaining the registered high selectivity for rotaxanes (Figure 13) [49].

## 5. Cyclopropanation Reactions Catalyzed by Metalloporphyrins

Of all the reactions using diazo compounds, cyclopropanations (Figure 1) have been the most studied as they simultaneously form two C−C bonds. Cyclopropanes are very useful building blocks in organic synthesis and the best method for their synthesis is, via one-pot reaction, the addition of carbene (generated by several methods) to an olefin. Versatility regarding olefins and diazo compounds is quite diverse, generating many different compounds [38,50,51,52,53].

There are reports in the literature that hydrophobic pockets can accelerate cyclopropanation reactions. For example, a hybrid system of cationic iron porphyrin and DNA accelerates the cyclopropanation reactions due to the concentration of reagents occupying hydrophobic spaces close to DNA [54]. Considering the premise that DNA acts similarly to a micelle, Roelfes and collaborators [55] studied the combination of cationic Fe-porphyrins with anionic surfactants, such as sodium dodecyl sulphate (SDS), in concentrations above their critical micelle concentrations in water (Figure 14). It was concluded that micellar catalysis with surfactants eliminates the need for organic solvents and accelerates the cyclopropanation reaction of *p*-methoxystyrene with ethyl diazoacetate (the carbene precursor) with iron porphyrins. Moreover, the catalytic cyclopropanation of other styrene derivatives with different diazoacetates was considerably accelerated in the presence of the cationic iron porphyrin bearing four *para*-*N*-methylpyridinium groups at the *meso*-positions. In most of the cases, without the addition of surfactant, the reaction yield was below 5%. Highlighted in Figure 14 are some examples and the best catalysts that can be used for these reactions. The acceleration of these micellar reactions, similar to the cyclopropanation reactions catalyzed by heme and DNA enzymes, suggests that the main effect is the increase in molarity within the hydrophobic cavities.

Asymmetric cyclopropanation reactions using chiral catalysts and catalyzed N−H insertion reactions in the presence of diazo compounds are well-established methodologies. However, reactions using organometallic diazo derivatives are still a challenge, as the influence of the metal complex on the course of the reaction with these diazo compounds is not well understood. Specifically, the influence of the ferrocene group is not known in these reactions, nor the best reaction conditions. To investigate the influence of the ferrocenyl metal complex, Simonneaux and collaborators [56] investigated (1) the asymmetric cyclopropanation of styrene derivatives with diazoacetylferrocene in the presence of the Halterman ruthenium chiral porphyrin (Figure 15); (2) N−H insertion of aminoesters with diazoacetylferrocene catalyzed by Fe(TPP)Cl (Figure 16).

In asymmetric cyclopropanation, the reaction between diazoacetylferrocene and different styrene was used to form the corresponding optically active ferrocenyl 2-arylcyclopropyl ketones (30–78% yield). The ferrocene group remained intact in the final cyclopropyl ketones, and the enantiomeric excesses found for the *trans*-isomer stayed between 74% and 96%. The N−H insertion reaction between diazoacetylferrocene and aminoesters was catalyzed by the tetraphenylporphyrin iron chloride Fe(TPP)Cl. The insertion is regioselective onto the NH_2_ group and occurs in high yields (82–87%), being chemoselective even in the presence of the O−H group of tyrosine.

Asymmetric cyclopropanation is one of the areas of the chemistry of diazo compounds that has evolved the most over the last few years through Rh and Ru-based catalysts [10,38,39]. However, there is always room to create new chiral catalysts and new reaction pathways to improve chiral induction under more favourable conditions [57]. Gallo and collaborators reported the synthesis of iron and ruthenium glycoporphyrins and their catalytic activity in cyclopropanation reactions by using diazo compounds as carbene precursors, thus concluding that the number and location of carbohydrate units (a cellobioside) on the porphyrin skeleton modulate the diastereoselectivity of the reactions. However, none of the complexes studied induced enantiocontrol, probably due to the long distance between the chiral carbohydrates and the active metal centre [58].

Zhang and collaborators [59] developed the asymmetric radical cyclopropanation of alkenes using *N*-arylsulfonyl hydrazones (as diazo precursors) to generate metalloradicals from chiral Co−porphyrins, followed by their insertion into the double bond of the alkene. This reaction produced cyclopropanes in high yields with effective control of diastereo- and enantioselectivity. Following this line of research, Zhang, and collaborators [57,60] also developed a highly asymmetric system for radical cyclopropanation with asymmetric diazomalonates. The asymmetric reaction of 1,1-cyclopropanediesters was quite effective with several types of alkenes. The study of the mechanism of this reaction indicated that it happens via metalloradical catalysis (MRC). These optically active 1,1-cyclopropanediesters are important chiral building blocks in organic syntheses. The catalysts for the reactions are chiral Co−porphyrins containing chiral groups in an arrangement with D2 symmetry. The products obtained from 1,1-cyclopropanation can react with alkenes with various functional groups (Figure 17).

The origin of enantioselectivity is due to the non-covalent interactions of the catalytic system, which is very efficient in various styrene derivatives, regardless of the electronic nature of the aromatic alkene used. The reaction yields were high (up to 99%), as diastereoselectivity (ranged up to 94:6 dr) and enantioselectivity (up to 97% ee). Increasing steric hindrance did not affect the enantioselectivity of the reaction, but it did affect the reaction yield and diastereoselectivity. Unsaturated substrates reacted well but under different catalytic conditions.

Entities that have cavities or cages can function as catalysts and carry out selective reactions inside them through non-covalent host-guest interactions, or the cavities can allow the encapsulation of catalysts, in some cases enhancing their reactivity. This is how most enzymes catalyze reactions in their cavities, and because of the supramolecular interactions of substrates within these cavities, they exhibit high specificity and efficiency [61]. Many natural products, such as chiral cyclodextrins, have different cavity sizes and catalyze reactions (e.g., β-cyclodextrin). Many of these supramolecular entities were designed and synthesized to mimic the structure and functionality of discrete coordination enzymes to accelerate substrate-specific reactions and manipulate regio- and enantioselectivity [62]. In this line of research, the calixarene macrocycle can be highlighted, which has phenolic units connected by methylene bridges in the ortho-positions relative to the hydroxyl group. These caged entities combine a hydrophobic cavity and a hydrophilic external surface that includes various substrates, accommodating metallic compounds and catalyzing reactions with results that surpass reactions with different types of catalysis [63].

Mouarrawis et al. were able to synthesize three cubic cages with different exopolarities attributed to the different groups on the periphery (Figure 3). These cages were used as hosts to encapsulate the catalytic active cobalt(II) *meso*-tetra(4-pyridyl)porphyrin, the guest. The resulting caged catalysts were studied in the cobalt-catalyzed cyclopropanation reaction of styrene with EDA (Figure 3) involving cobalt-carbene radical intermediates. The exofunctionalized cage catalyst with apolar icosyl groups evidenced higher activity than the non-functionalized or the polar (PEG-4) exofunctionalized counterparts. However, the polar PEG exofunctionalized catalyst evidenced higher selectivity for the cyclopropane product than the non-functionalized or the apolar icosyl exofunctionalized catalyst. On the other hand, encapsulation of the cobalt(II) *meso*-tetra(4-pyridyl)porphyrin guest into the cage with apolar icosyl groups led to a three times more active catalyst than Co(TPP) and a significantly increased TON if compared to both Co(TPP) and non-encapsulated cobalt(II) *meso*-tetra(4-pyridyl)porphyrin. According to the authors, the increased local concentration of the substrates in the hydrophobic cage compared to the bulk explains the higher catalytic activities registered [64,65].

Fischer and collaborators reported a catalytic process where the diastereoselectivity remarkably rests on specific local confinement effects, which can be modified by the careful choice of the MOF catalyst. The heterogeneous porphyrin-based MOF catalysts, PCN-222(Rh) and PCN-224(Rh) (Figure 4), that contain no stereocenters, were studied in the diastereoselective cyclopropanation of different styrenes with EDA, demonstrating good catalytic activity. When styrene and other non-coordinating olefins were the substrates, no diastereoselectivity was registered. Remarkably, styrenes carrying coordinating amino and hydroxy groups exhibited a high diastereomeric ratio (dr) of up to 23:1 (*trans*:*cis*) over PCN-224(Rh), which was attributed to substrate coordination to neighboring Rh centers caused by local cavity confinement effects. For 4-aminostyrene, the diastereoselectivity was increased to a dr of 42:1 (*trans*:*cis*) over PCN-222(Rh), a structural analogue of PCN-224(Rh), although featuring shorter Rh–Rh distances, that is 9.7 Å for PCN-222(Rh) and 13.6 Å for PCN-224(Rh) [66].

Olefin cyclopropanation was studied in the synthesis of rotaxanes by radical carbene transfer reactions promoted by Co(II) porphyrin-based semi-rigid macrocycle, and the highest yield (95%) was obtained in the presence of 3,5-diphenylpyridine as axial ligand. The active-metal-template strategy, which includes the radical-type activation of ligands by the cobalt ion of the porphyrin, is the basis for the reported methodology (Figure 18) [68,69].

## 6. Catalytic Properties of Metalloenzymes and Hemoproteins

Metalloporphyrins are recognized as useful supports for oxene, carbene, and nitrene transfer reactions. The development of artificial hemoproteins, which can be seen as non-natural oxene, carbene, and nitrene transferases, was inspired by natural heme monooxygenases. Hence, these activities were originally revealed by testing hemoproteins for their ability to mimic the identified activities of metalloporphyrin catalysts [70,71]. Being aware that the first evidence of hemoproteins acting as biocatalysts for carbene transfer reactions was published online on 12 December 2012 [72], we may realize why engineered metalloenzymes and hemoproteins (e.g., myoglobin, cytochrome P450s) only recently arose as a highly promising tool to create biocatalysts for a broad range of applications involving non-native substrates, namely those concerning carbene-transfer reactions [20,70,73,74,75,76,77,78,79,80,81,82,83,84,85,86].

One example is the cyclopropanation of 5-chloropent-1-ene with diazoacetone to produce the corresponding cyclopropyl ketone in the presence of hemoprotein biocatalysts derived from thermophilic bacterial globins (Figure 19a). These biocatalysts were developed as variants of the mutant heme protein derived from the thermophile *Methylacidophilium infernorum* (named Hell’s Gate globin I–HGG), reaching high diastereoselectivity (*cis*/*trans* ratio up to 1:99) and good enantioselectivity for (1*R*,2*R*) enantiomer (75% *ee*) [77].

In recent years, Fasan and co-workers have demonstrated the ability of engineered metalloenzymes for several efficient and selective biocatalytic transformations. The development of an iron-based biocatalyst for enantioselective α-C−H functionalization of pyrrolidines, via carbene transfer reaction with diazoacetone, is a possible strategy (Figure 19b). This transformation could be achieved in high yields, high catalytic activity, and high stereoselectivity (up to 99% *ee* and over 20,000 TON) using engineered variants of CYP119 from *Sulfolobus solfataricus* [87]. Furthermore, an engineered dehaloperoxidase (DHP) enzyme (from *Amphitrite ornata*) was used as carbene transferase for the stereoselective synthesis of cyclopropanol esters (with up to 99% *de* and *ee*) through the biocatalytic asymmetric cyclopropanation of vinyl esters with EDA (Figure 19c) [88].

More recently, an engineered artificial metalloenzyme containing an Ir–porphyrin complex was reported. In this work, *E. coli* cells expressing Ir–CYP119 catalyzed the cyclopropanation of (−)-limonene with high diastereoselectivity. So, by using a heterologous heme transport system, the authors constructed an artificial biosynthetic pathway incorporating an Ir-porphyrin-based metalloenzyme in *E. coli* (Figure 19d) [79]. The same research group described artificial metalloenzymes generated from the combination of a CYP450 scaffold and an Ir-porphyrin cofactor that catalyze the intermolecular insertion of carbenes into the C−H bonds of a range of phthalan derivatives containing substituents that render the two methylene positions in each phthalan nonequivalent (Figure 19e). These reactions occur with site selectivity ratios of up to 17.8:1 and, in most cases, with pairs of enzyme mutants that preferentially form each of the two constitutional isomers [89]. Finally, the authors have shown that the non-pathogenic *E. coli* strain Nissle 1917 (EcN), possessing a genetically encoded transport system, is a suitable host for the efficient uptake of an Ir-porphyrin complex and the in vivo assembly of Ir-CYP119. This strain enabled stereoselective and site-selective functionalization by carbene insertion into benzylic C−H bonds of phthalan derivatives catalyzed by an artificial metalloenzyme in whole cells. It was shown to accelerate the directed evolution of Ir-CYP119 by enabling high-throughput screening of reactions with new substrates in whole cells [90].

The intermolecular cyclopropanation reaction using a phosphorus-containing diazo compound [dimethyl(diazomethyl)phosphonate)] as carbene precursor, developed by Fasan and co-workers, was considered to be the first example of an efficient and enantioselective synthesis of enantioenriched cyclopropylphosphonate esters (up to 99% *de* and *ee* for the (1*S*,2*R*) stereoisomer), catalyzed by myoglobin-based biocatalysts (Mb variants; Figure 19f) [91]. Using also engineered myoglobin catalysts, the same research group reported the cyclopropanation of α-difluoromethyl alkenes in the presence of EDA, affording CHF_2_-containing cyclopropanes in high yield (up to >99%) and with high stereoselectivity (up to >99% *de* and *ee*) [92], along with the construction of 2,3-dihydrobenzofurans in high enantiopurity (>99.9% *de* and *ee*) and high yields via benzofuran cyclopropanation [93]. Additionally, a myoglobin-based carbene transferase incorporating a non-native iron-porphyrin cofactor and axial ligand showed to be highly efficient as a catalyst for the asymmetric cyclopropanation of electron-rich and electron-poor alkenes, reaching high diastereo- and enantioselectivity (up to >99% *de* and *ee*). Mechanistic studies showed that the reaction depends on radical-type carbene-transfer reactivity due to the reconfigured primary coordination sphere around the iron center [94].

Moreover, biocatalysts derived from sperm whale myoglobin proved to be active for carbene transfer in the asymmetric synthesis of fused cyclopropane-δ-lactones via intramolecular cyclopropanation of homoallylic diazoacetates in high yields and with up to 99% *ee* (Figure 19g) [95], and in the asymmetric synthesis of fused cyclopropane-γ-lactams via cyclization of allyl diazoacetamides into the corresponding bicyclic lactams, as can be seen in Figure 19h for (*E*)-2-diazo-*N*-(3-(4-fluorophenyl)allyl)-*N*-methylacetamide, in high yields and up to 99% *ee* [96].

Most recently, Arnold and co-workers reported that engineered hemoproteins derived from a bacterial cytochrome P450 can catalyze the synthesis of chiral 1,2,3-polysubstituted cyclopropanes, regardless of the stereo purity of the olefin substrates used. Cytochrome P450_BM3_ variant P411-INC-5185 exclusively converts (*Z*)-enol acetates into enantio- and diastereo-enriched cyclopropanes if starting from mixtures of (*Z*/*E*)-olefins. So, in the end, the reaction delivers a leftover (*E*)-enol acetate with 98% stereo purity (Figure 20a) [97].

This very recent work is just the extension of a path that began a decade ago in Professor F. Arnold’s research lab, which gave rise to a remarkable journey based on directed evolution, offering new meanings to the world of enzymatic catalysis, of which we highlight below some of the most recent discoveries. One example is the results obtained for the construction of C−C bonds through sp^3^ C–H functionalization, achieved by using *E. coli* expressing cytochrome P411-CHF iron-based catalyst, derived from a cytochrome P450 enzyme in which the native cysteine axial ligand was substituted by serine (cytochrome P411). This engineered iron-based catalyst evidenced enantio-, regio- and chemoselectivity for the intermolecular alkylation of sp^3^ C–H bonds through carbene C–H insertion (Figure 20b) [98]. On the other hand, the P411-HF variant stood out as a highly active alkylation enzyme catalyst for the alkylation of indoles, as exemplified for 1-methylindole (Figure 20c). Moreover, no N−H insertion products were observed, and alkylated products were isolated in good yields across a range of substituted, unprotected indoles, knowing how transferring carbene moieties to heterocycles to obtain C(sp^2^)−H alkylation products are valuable transformations in organic synthesis [99]. Likewise, by engineering cytochrome P450 enzymes, Arnold and co-workers were able to develop several P411 variants able to catalyze the insertion of fluoroalkyl carbenes into α-amino C(sp^3^)−H bonds, as shown for *N*-phenylpyrrolidine and 2,2,2-trifluoro-1-diazoethane. The enantiodivergent synthesis of fluoroalkyl-containing molecules turned out to be possible with P411-PFA and P411-PFA-(*S*) variants as biocatalysts, originating selectively the (*R*)- and the (*S*)-enantiomer, respectively (Figure 20d). Finally, these variants could install a trifluoroethyl group onto various *N*-aryl pyrrolidine substrates by directly activating the α-amino C−H bonds, thus achieving excellent turnovers and enantiomeric excess (*ee*) up to 99% [100]. Following the stereoselective carbene addition to terminal alkynes to produce cyclopropenes (P411-C6 variant as biocatalyst) and bicyclo [1.1.0]butanes (P411-E10 variant as biocatalyst) [101], the more challenging carbene transfer to internal alkynes for cyclopropene synthesis was achieved with P411-C10 variants (which belong to the family of P411-CHF) with impressive efficiency and stereoselectivity (all with >99.9% *ee*), as illustrated for 1-phenylbutyne and EDA (Figure 20e) [102]. This P411-C10 engineered enzyme was also shown to be efficient for lactone carbene insertion into primary and secondary α-amino C−H bonds, thus allowing chiral lactone derivatives synthesis with high catalytic efficiency. Moreover, for carbene insertion into secondary C−H bonds, a single mutation was uncovered to invert the two contiguous chiral centers, hence leading to the opposite enantiomers of the same major diastereomers, thus in a stereo divergent manner (Figure 20f) [103]. Based on their previous work, which demonstrated that variants of a heme protein, *Rhodothermus marinus* cytochrome *c* (*Rma cyt c*), catalyze abiological carbene boron–hydrogen (B–H) bond insertion with high efficiency and selectivity [104], the authors explored a similar approach with lactone-based carbenes. One of the *Rma cyt c* variants showed high selectivity and efficiency for B–H insertion of 5- and 6-membered lactone carbenes (Figure 20g) [104].

Engineered variants of *Aeropyrum pernix* protoglobin (*Ape*Pgb) represent the first example of a biocatalyst for carbene transfer from diazirines (cyclic isomers of diazo compounds) at ambient temperature, not requiring exogenous heat or light. Moderate yields and modest diastereo- or enatioselectivity values were reached, as exemplified for benzyl acrylate, using 3-phenyl-3*H*-diazirine as a carbene precursor (Figure 20h) [105]. The same *Ape*Pgb variant was shown to catalyze the cyclopropanation of unactivated alkenes using EDA, yielding the corresponding *cis*-cyclopropanes [106], which was the basis for the development of a method for the challenging synthesis of *cis*-trifluoromethyl-substituted cyclopropanes (Figure 20i) using *Ape*Pgb that can catalyze the reaction with low-to-excellent yield (6–55%) and enantioselectivity (17–99% *ee*), depending on the substrate [107]. The same research group reported the enantioselective one-carbon ring enlargement of aziridines into azetidines, where two new bonds are formed (one C−C and one C−N bond) through a [1,2]-Stevens rearrangement strategy, catalyzed by P411-AzetS, an engineered variant of cytochrome P450_BM3_, which exhibited carbene transferase activity with utmost stereo control, favoring the (*S*)-enantiomer (99:1 er) (Figure 20j) [108]. Very recently, a high-affinity heme-binding protein with an open coordination site adjacent to a large reconfigurable substrate binding cavity was designed from scratch, and its catalytic activity tested for the enantioselective cyclopropanation of styrene with EDA (up to 93% isolated yield, 5000 TON, 97:3 e.r.) [85].

## 7. Final Remarks

The chemistry of diazo compounds has always aroused the interest of synthetic organic chemists due to the many reactions that can be carried out through the decomposition of these compounds. Some of these reactions are very difficult to perform by other methods. These reactions proceed through the formation of carbenes or metallocarbenes, depending on the reaction conditions. Reactions catalyzed by low-cost metal-complexed porphyrins offer advantages over rarer transition metal-complexed porphyrins. Porphyrins complexed with these metals are powerful tools for creating new C−C, C−H, C=C, O−H, N−H, S−H, Se−H bonds, etc. Depending on the porphyrin system involved, the carbene insertion can be efficiently targeted to a specific functional group for the synthesis of a broad portfolio of fine chemicals. For this reason, in recent years, porphyrins have been highly efficient and low-cost catalysts. Particularly, those Fe and Co complexes have promoted the alkene cyclopropanations, C−H and X−H functionalizations (X = N, O, S, Se, Si, Sn, Ge).

Metalloenzyme-catalyzed carbene transformations are potent routes for creating tricky molecules. These lab-designed enzymes harness proteins’ ability to control reactive carbene species, ensuring precise outcomes. Novel artificial carbene transferases enable diverse methodologies, even unprecedented ones, which cover various reactions such as cyclopropanation, cyclopropenation, and carbene X–H insertion. In these conversions, biocatalysts surpass small-molecule catalysts in selectivity and turnover. The integration of these enzymatic reactions into synthesis, biological pathways, and chemo-enzymatic cascades is promising despite current limitations.

The new achievements involving diazo compounds discussed in this updated review and resulting from the studies carried out in the last five to six years provide an overview of the significance of such compounds in novel organic synthesis procedures.

## 8. Future Research Directions and Perspectives

The use of porphyrin-based catalysts in the decomposition of diazo compounds is an interesting research area with potential future directions and perspectives. Diazo compounds are versatile synthetic intermediates that can undergo various transformations, and their controlled decomposition is a crucial step in many organic synthesis processes. Several potential future research directions and perspectives can be highlighted in utilizing porphyrin-based catalysts for diazo compound decomposition, such as mechanistic insights and the pathways involved in the decomposition of diazo compounds catalyzed by porphyrin-based catalysts. Investigating reaction intermediates and transition states can provide insights into the factors influencing catalytic efficiency and selectivity, ligand design that coordinate to the metal core of porphyrins and could enhance the catalytic activity and selectivity in diazo compound decomposition. Ligand modifications can influence the electronic and steric properties of the catalyst, affecting its reactivity; substrate scope of porphyrin-based catalysts in diazo compound decomposition is essential and can provide valuable information about the catalyst’s versatility; enantioselective catalysis using chiral porphyrin-based catalysts is an emerging research direction and an opportunities for the synthesis of chiral building blocks and molecules; catalytic site engineering to explore site-specific modifications of the porphyrin catalyst’s active site can lead to enhanced catalytic properties; porphyrin-based catalysts can also be utilized in photocatalytic diazo compound decomposition reactions and sustainable catalysis that focus on the development of sustainable and environmentally friendly processes using porphyrin-based catalysts coupled with green solvents, mild reaction conditions, and renewable resources could be integrated into the catalytic systems.

## Data Availability

Not applicable.

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
