# Peer review of "Recent Synthetic Advances on the Use of Diazo Compounds Catalyzed by Metalloporphyrins"

_molecules, 2023, doi:10.3390/molecules28186683_

Round 1
Reviewer 1 Report
Reviewer #: This review has summarized the advancements in the investigations involving the field of diazo compounds catalyzed by metalloporphyrins (M-Porph, M = Fe, Ru, Os, Co, Rh, Ir). It highlights the importance of metalloporphyrins in a variety of organic transformations involving diazo compounds which are organic substances that are often used as precursors in organic synthesis.
1) A review article aims at achieving THREE essential objectives:
Objective i) providing a complete, structured and systematic summarization on the related key aspects. This means that the authors will summarize those in many figures and tables;
Objectives 1) has been partially achieved, as some of the key aspects of the field is not covered in the present review. Such as, a section about the preparation of corresponding metalloporphyrins and their important characterization technique would offer valuable information to the readers. There is not even a single table provided in the review, a table would offer comprehensive knowledge about the field in one place.
Objective ii) presenting new discoveries from the authors’ own knowledge synthesis based on existing literature results. This means that the authors will provide important and synthesized new knowledge that are not included in those articles in the literature;
Objective ii) has only been partially achieved, the authors' own critical thinking via comparing/benchmarking the literature for new knowledge synthesis is not there in the review.
Objective iii) outlining detailed views on future research directions and perspectives.
Objective iii) has been partially achieved (see below).
Therefore, the authors are strongly suggested to revise the review paper to address these weaknesses before the review paper is suitable for publication.as well as important concepts and difficulties in this area together with mechanisms/reaction pathways utilizing various substrates.
1) The numbering of sections and subsections are problematic and must be rectified. A index must be provided for the readers
2) At the end of “Introduction” section, the authors need to justify why the review article fills a critical gap in the field, is indeed in need and timely. To my knowledge, Molecules” or any other high impact journals have already published many review articles related to the topics covered in current review paper. Please go through those review articles and justify what make your review paper worthwhile for publication?
4) This is a review paper that should be forwarding-looking. At present, future research perspectives are only briefly lumped in the last section "Conclusions and perspective". To make the review article be more forward-looking, please separate this section into two sections, i.e. a section with title like "Challenges and Perspectives" and a section with title like "Conclusions". The new section "Challenges and Perspectives" should be expanded to systematically outline the authors' detailed views on the key challenges, existing research gaps and future research directions etc., including also potential future developments in research methodologies (both experimental and modelling) that are important to future development of the related field. This section brings critical benefits to the readership and usually have at least 1 page in printing form.
The new section "Conclusions" can then wrap up your review paper with concluding remarks.
see the report
Author Response
Manuscript ID: molecules-2543575
Title: "Recent synthetic advances on the use of diazo compounds catalyzed by metalloporphyrins"
Authors: Mário M. Q. Simões, José A. S. Cavaleiro and Vítor F. Ferreira
Dear Editor,
Please find below our answers to the reviewers #1.
Referee #1:
Referee 1 has his own ideas on how to present a review, which are detailed below:
1) providing a complete, structured and systematic summarization on the related key aspects; 2) presenting new discoveries from the authors’ own knowledge synthesis based on existing literature results; 3) outlining detailed views on future research directions and perspectives.
Authors comments:
The authors of the present manuscript, who have already published many reviews and book chapters, have contrary opinions and respectfully disagree with referee 1. The disagreements are the following: there is no established general rule for reviews; authors have publications in the area that give them capacity to write on the topic; and a third one on whether the review follows an important theme in the literature, we have to state that our manuscript covers a gap in the literature since it is a complete review of publications involving the specified topic during the last 6 years, as outlined in its objectives.
We also do not agree with the reviewer in considering a section on the synthesis of porphyrins in the review, since most catalysts based on porphyrins were commercially available or were carried out by traditional synthetic methods already included in several publications and even in several reviews of the authors of the present manuscript. A main objective of the review would be related with the fact that porphyrins can be efficient catalysts for decomposition reactions of diazocompounds, bringing in such way, fantastic potentialities in organic synthesis. But the synthesis of porphyrins, we mean simple porphyrins, was not a target for the present manuscript.
About the question: “The numbering of sections and subsections are problematic and must be rectified. An index must be provided for the readers”.
Answer. We have to say that our submitted manuscript has the numbering of the sections as required by the standards and the template. We do not feel it necessary to include subsections. The Molecules template was followed and there is no request for index. We also verified this situation in 6 of the most recent revisions published in Molecules in which the “index” item does not appear (Molecules 2023, 28, 5614; Molecules 2023, 28, 5610; Molecules 2023, 28, 5604; Molecules 2023, 28, 5588; Molecules 2023, 28, 5586; Molecules 2023, 28, 5576).
About the question: “At the end of “Introduction” section, the authors need to justify why the review article fills a critical gap in the field, is indeed in need and timely. To my knowledge, Molecules” or any other high impact journals have already published many review articles related to the topics covered in current review paper. Please go through those review articles and justify what make your review paper worthwhile for publication?”
Answer: In 2018 the authors published the article “Carbene Transfer Reactions Catalyzed by Dyes of the Metalloporphyrin Group”, by Mário M. Q. Simões et al., Molecules 2018, 23(4), 792 which covered the literature until the year 2017. Since then, no other review has been published covering the gap of the following years. The initial survey showed that there were many new publications on the subject at hand. Therefore, we decided to update what has been published on the catalytic decomposition of diazo compounds with metalloporphyrins. All significant revisions in that period, and even earlier ones that were pertinent, were cited.
About the question: This is a review paper that should be forwarding-looking. At present, future research perspectives are only briefly lumped in the last section "Conclusions and perspective". To make the review article be more forward-looking, please separate this section into two sections, i.e. a section with title like "Challenges and Perspectives" and a section with title like "Conclusions". The new section "Challenges and Perspectives" should be expanded to systematically outline the authors' detailed views on the key challenges, existing research gaps and future research directions etc., including also potential future developments in research methodologies (both experimental and modelling) that are important to future development of the related field. This section brings critical benefits to the readership and usually have at least 1 page in printing form. The new section "Conclusions" can then wrap up your review paper with concluding remarks.
Answer: as it can be seen in the “Final Remarks” of section 6, several predictions were presented that can be understood as perspectives, such as: “The chemistry of diazo compounds has always aroused the interest of synthetic organic chemists due to the many reactions that can be carried out through the decomposition of these compounds. Some of these reactions are very difficult to perform by other methods.” and “The new achievements involving diazo compounds and porphyrin complexes discussed in this updated review and resulting from the studies carried out in the last 5 to 6 years, provide an overview on the significance of such compounds in novel organic synthesis procedures.”
“6. Final Remarks: The chemistry of diazo compounds has always aroused the interest of synthetic organic chemists due to the many reactions that can be carried out through the decomposition of these compounds. Some of these reactions are very difficult to perform by other methods. These reactions proceed through the formation of carbenes or metallocarbenes, depending on the reaction conditions. Reactions catalyzed by low-cost metal-complexed porphyrins offer advantages over rarer transition metal-complexed porphyrins. Porphyrins complexed with these metals are powerful tools for creating new C-C, C-H, C=C, O-H, N-H, S-H, Se-H bonds, etc. Depending on the porphyrin system involved, the carbene insertion can be efficiently targeted to a specific functional group for the synthesis of a broad portfolio of fine chemicals. For this reason, in recent years, porphyrins have been highly efficient and low-cost catalysts. Particularly those which are Fe and Co complexes have promoted the alkene cyclopropanations, C-H and X-H functionalizations (X = N, O, S, Se, Si, Sn, Ge). The new achievements involving diazo compounds and porphyrin complexes discussed in this updated review and resulting from the studies carried out in the last 5 to 6 years, provide an overview on the significance of such compounds in novel organic synthesis procedures.”
Although we understand the point of view that referee #1 argues, in our conception the review article should present what was done in the time gap for the theme that was proposed in the study. Moreover, Molecules does not require a "Conclusions and Perspectives" section.
We hope that the new version is now suitable to be published in Molecules.
Thank you in advance.
Sincerely yours,
Professor Vítor F. Ferreira
Departamento de Tecnologia Farmacêutica Química, Universidade Federal Fluminense, 24241‑002, Niterói, RJ, Brazil
Reviewer 2 Report
Manuscript of Ferreira with co-workers is a review of the activation of diazo compounds in organic synthesis catalyzed by metalloporphyrins based on recent works (5 years). The manuscript is well written, and with no doubt, it deserves publication in Molecules after response to the following minor remarks:
1) The use of the term “carbenoid” (p. 1-2, and Scheme 1) in the case of a carbene transfer from a diazo compound is wrong. See Chem Eur J 2017, 23, 14389-14393 (doi: 10.1002/chem.201702392). “Metal–carbene species” or “Metal–carbene complexes” is better.
2) The important previous review in this field (Chem. Soc. Rev., 2011,40, 1950-1975, doi: 10.1039/C0CS00142B) should be cited.
3) In Scheme 6, “AgSBF6” is incorrectly written. Should be AgSbF6.
4) In Scheme 10, general formula for gem-alkene is given, however, no one example of such alkenes is given in the substrate scope.
5) Lines 144-146 and 302-304. Mentions of the use of Rh and Ru complexes should be supported by literature references (at least some reviews).
Author Response
Manuscript ID: molecules-2543575
Title: "Recent synthetic advances on the use of diazo compounds catalyzed by metalloporphyrins"
Authors: Mário M. Q. Simões, José A. S. Cavaleiro and Vítor F. Ferreira
Dear Editor,
Please find below our answers to the reviewers #2.
Referee #2:
Referee 2 has suggested minor corrections, which are detailed below:
About the question: “1) The use of the term “carbenoid” (p. 1-2, and Scheme 1) in the case of a carbene transfer from a diazo compound is wrong. See Chem Eur J 2017, 23, 14389-14393 (doi: 10.1002/chem.201702392). “Metal–carbene species” or “Metal–carbene complexes” is better.”
Answer. The authors wish to acknowledge referee 2 for this information. We have changed the term “carbenoid” into “metal–carbene complexes” in the manuscript (highlighted in the text using the yellow color).
About the question: “2) The important previous review in this field (Chem. Soc. Rev., 2011,40, 1950-1975, doi: 10.1039/C0CS00142B) should be cited.”
Answer. The authors also recognize this valuable suggestion. We have introduced the suggested review in the manuscript [New Reference: #32].
Note: this review (Chem. Soc. Rev., 2011,40, 1950-1975) has been cited as REF. #33 in our previous review (Molecules, 2018, 23, 792), entitled “Carbene Transfer Reactions Catalysed by Dyes of the Metalloporphyrin Group”.
About the question: “3) In Scheme 6, “AgSBF6” is incorrectly written. Should be AgSbF6.”
Answer. The authors wish to thank referee 2 for the correction. We have changed Scheme 6 accordingly.
About the question: “4) In Scheme 10, general formula for gem-alkene is given, however, no one example of such alkenes is given in the substrate scope.”
Answer. The authors are grateful for this proposal. Scheme 10 was modified, and one example was added.
About the question: “5) Lines 144-146 and 302-304. Mentions of the use of Rh and Ru complexes should be supported by literature references (at least some reviews).”
Answer. Appropriate references were carefully introduced in the appropriate place in the manuscript (highlighted in the text using the yellow color).
We hope that the new version is now suitable to be published in Molecules.
Thank you in advance.
Sincerely yours,
Professor Vítor F. Ferreira
Departamento de Tecnologia Farmacêutica Química, Universidade Federal Fluminense, 24241‑002, Niterói, RJ, Brazil